# Development and Evaluation of the Ultrasonic Welding Process for Copper-Aluminium Dissimilar Welding †

Rafael Gomes Nunes Silva [1,*], Sylvia De Meester [2], Koen Faes [1] and Wim De Waele [2]

1 Belgian Welding Institute, 9050 Ghent, Belgium; koen.faes@bil-ibs.be
2 Department of Electromechanical, Systems and Material Engineering, Ghent University, 9050 Ghent, Belgium; sylvia.demeester@gmail.com (S.D.M.); wim.dewaele@ugent.be (W.D.W.)
* Correspondence: rafael.nunes@bil-ibs.be
† This paper is an extended version of our paper "Development and Evaluation of the Ultrasonic Welding Process of Copper-Aluminium Dissimilar Materials" published in Proceedings of the 2nd International Conference on Advanced Joining Processes, Sintra, Portugal, 21–22 October 2021.

**Abstract:** The demand for joining dissimilar metals has exponentially increased due to the global concerns about climate change, especially for electric vehicles in the automotive industry. Ultrasonic welding (USW) surges as a very promising technique to join dissimilar metals, providing strength and electric conductivity, in addition to avoid metallurgical defects, such as the formation of intermetallic compounds, brittle phases and porosities. However, USW is a very sensitive process, which depends on many parameters. This work evaluates the impact of the process parameters on the quality of ultrasonic spot welds between copper and aluminium plates. The weld quality is assessed based on the tensile strength of the joints and metallographic examination of the weld cross-sections. Furthermore, the welding energy is examined for the different welding conditions. This is done to evaluate the influence of each parameter on the heat input resulting from friction at the weld interface and on the weld quality. From the obtained results, it was possible to optimise parameters to achieve satisfactory weld quality in 1.0 mm thick Al–Cu plate joints in terms of mechanical and metallurgical properties.

**Keywords:** Cu–Al welding; metallographic examination; parameter optimisation; tensile strength; USW

## 1. Introduction

Although the ultrasonic welding (USW) process has been used in academic and industrial applications for several decades, the recent uptake of green technology in the automotive industry has made the solid-state process a hot research topic again. Electric vehicles, hybrid or plug-in hybrid electric vehicles are increasingly being used for the reduction of emission of greenhouse gases and meeting national and international standards and legislation of emission targets [1]. Within the new challenges of electric vehicles manufacturing, the joining of dissimilar metals such as aluminium to copper appears as a key process for the manufacturing of battery packs [2–7], where strength and electrical conductivity are extremely important criteria for a satisfactory connection [8]. The joint strength and integrity should be sufficient to withstand all the impact and vibrational forces [3,9].

Joining dissimilar metals with fusion welding techniques has always been very challenging due to differences in melting temperature, the formation of brittle intermetallic compounds (IMCs) and the sensitive mechanical properties of the welded materials [6–10]. In addition, fusion welding processes may not be suitable for highly conductive and reflective metals when a large welding nugget is expected [11]. As a solution to these problems, solid-state welding has become popular due to the elimination of metallurgical defects such as the formation of IMCs, brittle phases and porosities in the fused zone as a result of liquid phases reactions [12–14].

In this scenario, USW surges as a very promising technique to join dissimilar metals for electric automotive battery manufacturing [9,15,16]. As a solid-state welding process, USW avoids melting of the materials and joins them based on diffusion and adhesion of the softened metals due to interfacial friction [3,17]. USW seems advantageous for the mentioned applications, as it provides the necessary joint strength and offers low or no brittle intermetallic layers along the weld line, which ensures less electrical resistance [9,18,19]. Hence, this process is suitable for highly conductive and reflective soft metals such as aluminium, copper, brass, silver and gold [20]. In conclusion, USW emerges as an appropriate technique for welding thin sheets applicable to various electric vehicle battery, electrical and electronics industries.

In the USW system, a piezoelectric transducer converts electrical energy into the shear vibration of a sonotrode, which causes the samples to be bonded together using a clamping force [21,22]. The oxide layers on the surface are removed from the plates interface [23,24], and the material is softened by the temperature rise at the specimen interface. The ultrasonic vibration leads to diffusion of the metals and, subsequently, adhesion [20]. Several bonding mechanisms have been reported in the literature including interfacial diffusion, adhesion by plastic deformation, local heating and mechanical interlocking [25].

Several researchers have investigated the welding mechanism present in the USW process for different applications, including joining of metal or non-metal sheets, metal–ceramic and metal–glass [9,17–19,26–29]. However, few studies were conducted considering Al–Cu dissimilar metal joining [8]. Among them, Satpathy and Sahoo [15], Zhao et al. [13] and Balasundaram et al. [30] evaluated the welding mechanism and micro-hardness distribution. Wu et al. [13] extended their study for a multi-layered Al–Cu joint and investigated the weld formation mechanism and failure modes using the lap shear test. Dhara and Das [11] evaluated the ultrasonic welding applied to three layers of Al sheet welded to a single layer of Cu sheet, investigating the welding mechanism, interfacial material mixing and micro-bond by optical microscopy, micro-hardness, as well as grain size and dynamic recrystallization using high-resolution EBSD Euler maps.

In this study, the authors propose a strategy for parametrization of USW dissimilar metal joining of single layers of Al and Cu, followed by the evaluation of each parameter and their combination on the welding energy and tensile strength of the lap shear specimen. Besides tensile strength, the welds are evaluated through peel testing, metallographic examination by optical microscope and EDX line scan mapping.

## 2. Materials and Methods

The experiments were performed on aluminium (EN AW-1050 H14) and copper (EN Cu-ETP) base materials. Electrolytic Tough Pitch (ETP) copper is an electrolytic refined copper that is commonly used in electrical applications due to its excellent electrical conductivity, thermal conductivity, ductility and corrosion resistance. The composition of the copper plates consists of Cu and O with a minimum value of 99.90% Cu and a maximum O content of 400 ppm. The measured percentage of Cu for the plates is 99.971%. The EN AW-1050 H14 is a popular grade of aluminium for general sheet work where moderate strength is required. The measured chemical composition of the aluminium plates is given in Table 1. All sample surfaces were cleaned before the welding process using acetone, following positive results achieved in exploratory welding trials.

**Table 1.** Chemical composition of EN AW-1050 H14.

| Al (%) | Si (%) | Fe (%) | Cu (%) | Mn (%) | Mg (%) | Ti (%) | Zn (%) | Cr + 3 (ppm) | Pb (ppm) |
|---|---|---|---|---|---|---|---|---|---|
| 99.549 | 0.091 | 2.269 | 0.011 | 0.009 | 0.008 | 0.008 | 0.017 | 2 | 1 |

The joint configuration and sheet dimensions are illustrated in Figure 1. The aluminium sheet was always used as the bottom part of the joint.

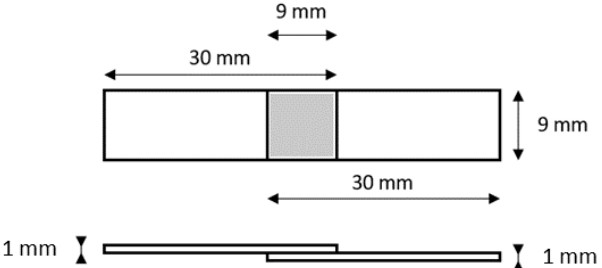

**Figure 1.** Joint configuration and sheet dimensions.

The ultrasonic welding machine used in the experiments was the Telsonic MPX Ultrasonics Linear Metal Welding Press, with a nominal power of 3.6 kW and a frequency of 20 kHz. The maximum load that can be reached is 1600 N, and the maximum vibration amplitude is 66 μm. The control software allows setting a vibration amplitude between 50% and 100% of this maximum value. This machine is suitable for welding both ferrous and non-ferrous alloys. The sonotrode moves horizontally back and forth, resulting in a tangential transfer of the waves to the workpieces.

All experiments are performed in time mode, as this mode tends to vary less than other modes and will therefore yield a better reference for comparison. The welding time, pressure and amplitude values are varied for each weld. The trigger time is fixed to 0.04 s, and the pressure build up time to 0.30 s. This means that the sonotrode starts to vibrate after 0.04 s until it reaches the required pressure after 0.30 s. During the experiments, the hold function is used to strengthen the weld after the weld cycle. The hold time and pressure are chosen equal to the selected time and pressure of the weld cycle.

For the metallographic examination, the optical microscope Olympus MX51 is used. To enable the evaluation of the weld cross-sections, the samples are cut perpendicular to the direction of the sonotrode vibration. After grinding and polishing, a first evaluation of the welds is performed. During this evaluation, the samples are inspected at different magnifications, starting at 12.5× and 50×, to determine the amount of welded area and to detect any potential weld imperfections. The areas of interest are then inspected in more detail using a larger magnification, such as 100×, 200× or even 500×. A second evaluation is done after etching the copper for 20 s with a solution of 10% ammonia in water saturated with hydrogen peroxide.

In addition to optical microscopy, SEM (scanning electron microscopy) images and EDX (energy-dispersive X-ray spectroscopy) mapping of the energy electrons are used to reveal information about the morphology of the welds.

In order to establish the window of suitable welding parameters, a series of exploratory tests was performed to determine the boundary conditions of each parameter. These welds were evaluated through visual and peel test examinations and classified in three levels based on the peel test results, as illustrated in Figure 2. The scoring of the welds was done according to levels 1, 3 and 5 to allow a quantitative comparison.

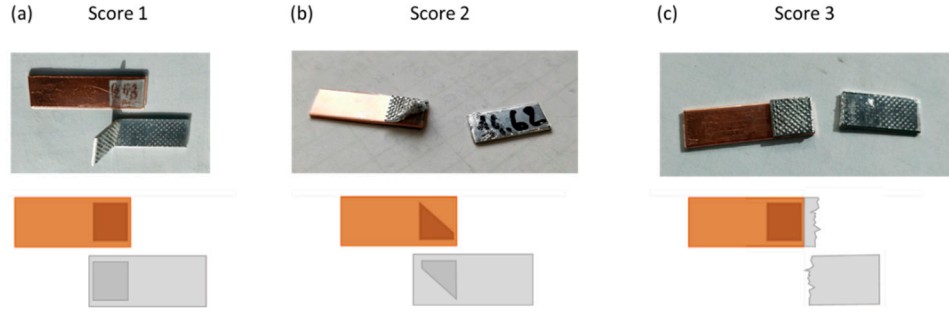

**Figure 2.** Peel test classification based on the type of joint failure, following the criteria for (**a**) low, (**b**) intermediate, and (**c**) high weld quality.

After the boundary conditions were determined, a full factorial Design of Experiments (DoE) was created, evaluating three welding parameters (welding time, pressure and vibration amplitude) at three levels, resulting in 27 weld tests, as shown in Figure 3. The values of the parameters corresponding with the points of the DoE matrix are listed in Table 2. To ensure the reproducibility of the process and results, four replicas were carried out. Among the replicas, three were used for tensile testing and one for metallographic examination.

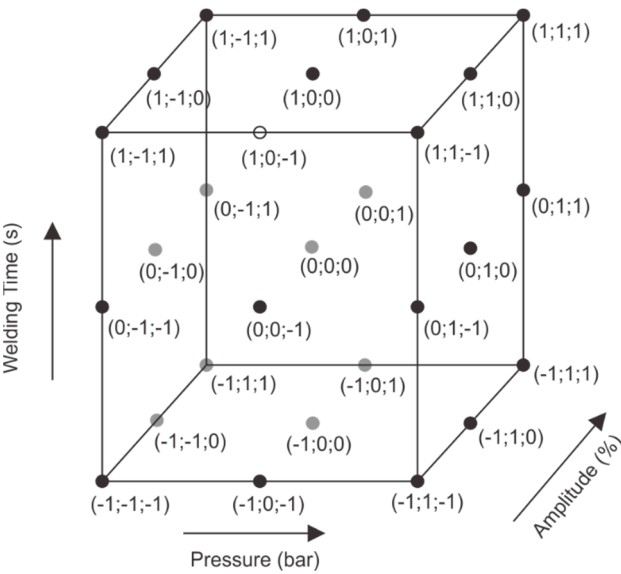

**Figure 3.** The values of the parameters corresponding with the points of the DoE matrix are listed in Table 2.

**Table 2.** Welding parameters used in the DoE matrix. The background color indicates the correlation of the value with the DoE level $(-1;0;1)$.

| DoE Level | Welding Time (s) | Pressure (bar) | Amplitude (%) |
|-----------|------------------|----------------|---------------|
| −1        | 0.8              | 2.2            | 75            |
| 0         | 1.5              | 2.7            | 80            |
| 1         | 2.2              | 3.2            | 85            |

The tensile strength of the welded specimens is determined using an Instron 8872 universal tensile test machine. The testing machine has a dynamic load capacity of 25 kN and uses a controlled displacement rate of 0.0333 mm/s. The tests are performed according to the EN ISO 14273 standard [31] for tensile shear testing of spot welds in overlapping sheets.

With the four welding energy values and three tensile tests for each condition, the standard deviation of the results has been calculated using Equation (1), with $\mu$ being the average value, $N$ the number of measurements and $x$ the measured energy value.

$$\sigma = \sqrt{\frac{1}{N} \sum_{i=1}^{N} (x_i - \mu)^2} \tag{1}$$

## 3. Results

### 3.1. Influence of the Welding Parameters on the Welding Energy

In the ultrasonic welding process, the welding energy is determined by the interaction of the above-mentioned welding parameters. Even knowing the exact conditions under which the weld tests were performed, it is also necessary to know the energy dissipation

and efficiency factor of the welding machine. With this uncertainty in mind, the energy value directly delivered by the welding machine after each weld cycle was assumed. The welding energy of all experiments and their respective mean and standard deviation values are shown in Table 3. As mentioned in the previous section, four replicas were made for each welding condition. The values of Energy A, B, C and D are related to those replicas.

**Table 3.** Welding energy of all performed welding tests, with their mean and standard deviation values. The background indicates the correlation of the value with the DoE level (−1;0;1).

| Weld Time (s) | Pressure (bar) | Amplitude (%) | Energy A (J) | Energy B (J) | Energy C (J) | Energy D (J) | Energy AVG (J) | Energy Std Dev (J) |
|---|---|---|---|---|---|---|---|---|
| 0.8 | 2.2 | 75 | 595.4 | 622.5 | 667.2 | 619.6 | 626.2 | 622.5 |
| 0.8 | 2.2 | 80 | 653.9 | 573.6 | 637.7 | 662.6 | 632.0 | 573.6 |
| 0.8 | 2.2 | 85 | 670.1 | 647 | 693.8 | 575.2 | 646.5 | 647 |
| 0.8 | 2.7 | 75 | 708.8 | 686.5 | 679.2 | 681.9 | 689.1 | 686.5 |
| 0.8 | 2.7 | 80 | 708.1 | 715.6 | 740.9 | 713.5 | 719.5 | 715.6 |
| 0.8 | 2.7 | 85 | 763.4 | 768.5 | 629 | 755.9 | 729.2 | 768.5 |
| 0.8 | 3.2 | 75 | 797.7 | 788.8 | 795.6 | 806.9 | 797.3 | 788.8 |
| 0.8 | 3.2 | 80 | 835.6 | 808.8 | 804.4 | 796 | 811.2 | 808.8 |
| 0.8 | 3.2 | 85 | 838 | 840.2 | 855.1 | 841.9 | 843.8 | 840.2 |
| 2.2 | 2.2 | 75 | 1591.4 | 1568 | 1533.6 | 1550.6 | 1560.9 | 1568 |
| 2.2 | 2.2 | 80 | 1548.1 | 1675.9 | 1689.9 | 1677.1 | 1647.8 | 1675.9 |
| 2.2 | 2.2 | 85 | 1721.3 | 1754.1 | 1739 | 1779 | 1748.4 | 1754.1 |
| 2.2 | 2.7 | 75 | 1922 | 1860.7 | 1872.6 | 1869.7 | 1881.3 | 1860.7 |
| 2.2 | 2.7 | 80 | 2042.9 | 2078.9 | 2030.5 | 1951.6 | 2026.0 | 2078.9 |
| 2.2 | 2.7 | 85 | 2102.6 | 2134.9 | 2003.1 | 2023.8 | 2066.1 | 2134.9 |
| 2.2 | 3.2 | 75 | 2175.4 | 2007.6 | 2058.6 | 2129.8 | 2092.9 | 2007.6 |
| 2.2 | 3.2 | 80 | 2380.3 | 2310.1 | 2293 | 2237.2 | 2305.2 | 2310.1 |
| 2.2 | 3.2 | 85 | 2391.1 | 2306.6 | 2443.1 | 2488.5 | 2407.3 | 2306.6 |
| 1.5 | 2.2 | 75 | 1131 | 1125.5 | 1144.1 | 1159.5 | 1140.0 | 1125.5 |
| 1.5 | 2.2 | 80 | 1170 | 1138.2 | 1061.2 | 1145.8 | 1128.8 | 1138.2 |
| 1.5 | 2.2 | 85 | 1183.7 | 1184.2 | 1205.6 | 1217.9 | 1197.9 | 1184.2 |
| 1.5 | 2.7 | 75 | 1301 | 1321.9 | 1264.6 | 1237.2 | 1281.2 | 1321.9 |
| 1.5 | 2.7 | 80 | 1389.6 | 1390.3 | 1344.7 | 1318.9 | 1360.9 | 1390.3 |
| 1.5 | 2.7 | 85 | 1473.1 | 1460.5 | 1402 | 1353.1 | 1422.2 | 1460.5 |
| 1.5 | 3.2 | 75 | 1499.9 | 1524.8 | 1455.4 | 1446.6 | 1481.7 | 1524.8 |
| 1.5 | 3.2 | 80 | 1613.6 | 1567 | 1606.6 | 1490.3 | 1569.4 | 1567 |
| 1.5 | 3.2 | 85 | 1671.6 | 1671.2 | 1640.8 | 1597.5 | 1645.3 | 1671.2 |

To evaluate the influence of each welding parameter and the interactions between them, a pareto chart was made. This chart is shown in Figure 4 and provides the standardized effects of the welding parameters on the average welding energy. The regression equation corresponding to these results is shown in Equation (2):

$$E = 339 + 507\,\mathrm{WT} + 112\,\mathrm{P} - 0.2\,\mathrm{A} - 266\,\mathrm{WT} \times \mathrm{P} - 5.6\,\mathrm{WT} \times \mathrm{A} - 2.17\,\mathrm{P} \times \mathrm{A} \\ + 7.2\,\mathrm{WT} \times \mathrm{P} \times \mathrm{A} \tag{2}$$

where E is the average energy (in J), WT is the welding time (in s), P is the pressure (in bar) and A is the amplitude (in %). This equation shows that the relative influence of the value of the welding time on the welding energy is very high. This can also be observed on the

Pareto chart, Figure 4, where clearly the standardized effect of the welding time is much larger than the effects of the other parameters and of the interactions of the parameters.

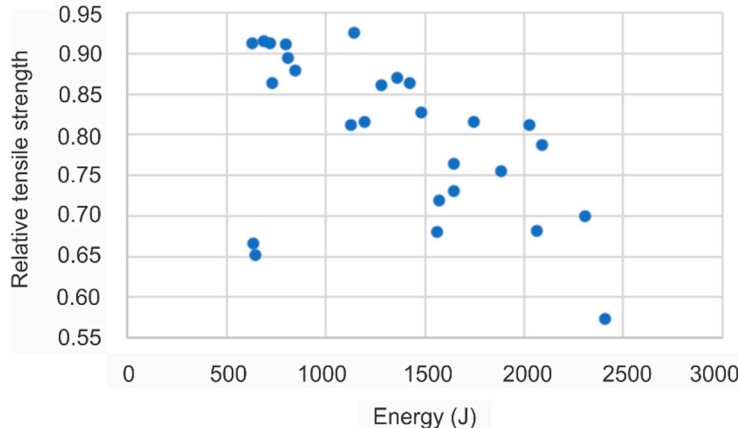

**Figure 4.** Pareto chart of the standardized effects of the welding parameters and their interactions on the average welding energy.

To avoid thermal degradation and softening of the materials due to a large heat input, the welding time should be kept rather low because the heat input is directly related to the welding energy. This also explains the low values of the welding time in the parameter window.

Contour plots of the average energy versus the amplitude and pressure for a constant welding time of 0.8 s, 1.5 s and 2.2 s are shown in Figure 5. In all three graphs, the maximal energy is indicated in dark green. The dark green areas all correspond to the regions where both the vibration amplitude and the pressure are maximal. As can be seen on the scale bar of the energy, an increase of the welding time results in much larger welding energy.

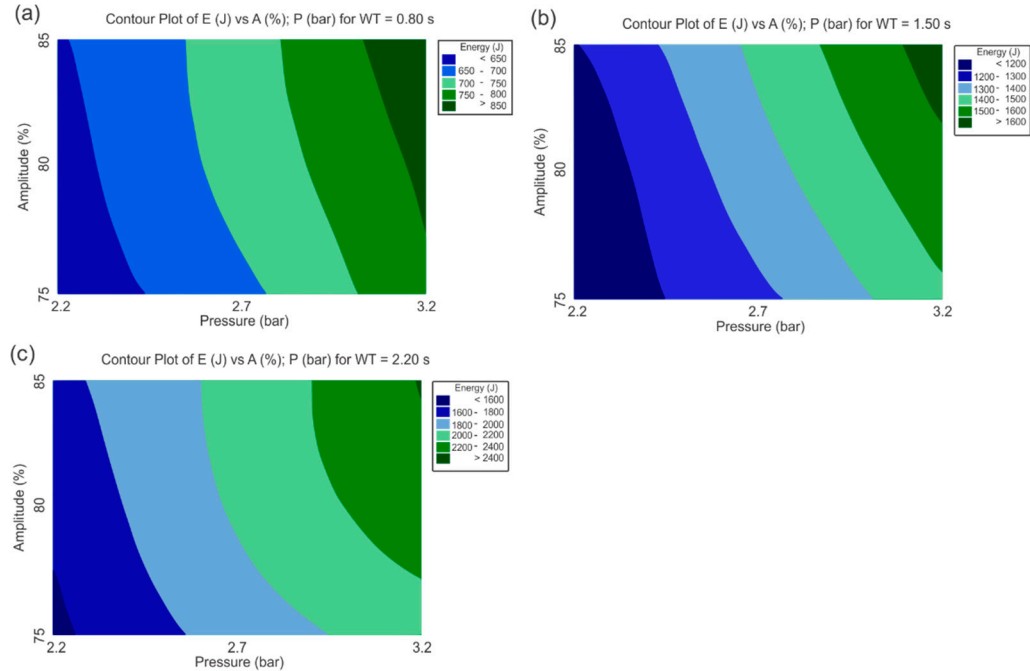

**Figure 5.** Contour plots of the average welding energy E vs amplitude A and pressure P at constant welding time WT of (**a**) 0.8 s, (**b**) 1.5 s and (**c**) 2.2 s.

### 3.2. Influence of the Welding Parameters on the Tensile Strength

To evaluate the joint quality of the aluminium–copper welds, tensile tests were performed on three replicas for each welding condition of the DoE. The resulting breaking force (F) for each weld is listed in Table 4. As replica A was used for metallographic examination, the three remaining replicas (B, C and D) were used for tensile testing. The tensile strength S can be calculated by dividing the maximum force by the nominal overlap area of the welds (Φ), as shown in Equation (3).

$$F = S \times \Phi \tag{3}$$

In this case, Φ equals 9.0 mm$^2$.

**Table 4.** Tensile strength of all welds, with their respective mean and standard deviation values. The background color indicates the correlation of the value with the DoE level (−1;0;1).

| Weld Time (s) | Pressure (bar) | Amplitude (%) | Tensile B (N) | Tensile C (N) | Tensile D (N) | Tensile AVG (N) | Tensile Std Dev (N) |
|---|---|---|---|---|---|---|---|
| 0.8 | 2.2 | 75 | 977 | 897 | 932 | 935 | 33 |
| 0.8 | 2.2 | 80 | 297 | 824 | 925 | 682 | 275 |
| 0.8 | 2.2 | 85 | 925 | 861 | 219 | 668 | 319 |
| 0.8 | 2.7 | 75 | 912 | 950 | 952 | 938 | 18 |
| 0.8 | 2.7 | 80 | 950 | 897 | 959 | 935 | 27 |
| 0.8 | 2.7 | 85 | 909 | 846 | 899 | 885 | 28 |
| 0.8 | 3.2 | 75 | 970 | 875 | 957 | 934 | 42 |
| 0.8 | 3.2 | 80 | 954 | 937 | 859 | 917 | 41 |
| 0.8 | 3.2 | 85 | 933 | 949 | 820 | 901 | 57 |
| 2.2 | 2.2 | 75 | 778 | 614 | 698 | 697 | 67 |
| 2.2 | 2.2 | 80 | 809 | 812 | 727 | 783 | 39 |
| 2.2 | 2.2 | 85 | 820 | 896 | 790 | 835 | 45 |
| 2.2 | 2.7 | 75 | 823 | 555 | 941 | 773 | 162 |
| 2.2 | 2.7 | 80 | 803 | 873 | 818 | 831 | 30 |
| 2.2 | 2.7 | 85 | 603 | 803 | 690 | 699 | 82 |
| 2.2 | 3.2 | 75 | 871 | 776 | 771 | 806 | 46 |
| 2.2 | 3.2 | 80 | 617 | 728 | 807 | 717 | 78 |
| 2.2 | 3.2 | 85 | 723 | 563 | 474 | 587 | 103 |
| 1.5 | 2.2 | 75 | 924 | 940 | 979 | 948 | 23 |
| 1.5 | 2.2 | 80 | 855 | 766 | 873 | 831 | 47 |
| 1.5 | 2.2 | 85 | 850 | 806 | 850 | 835 | 21 |
| 1.5 | 2.7 | 75 | 886 | 793 | 968 | 882 | 71 |
| 1.5 | 2.7 | 80 | 869 | 863 | 942 | 891 | 36 |
| 1.5 | 2.7 | 85 | 877 | 873 | 904 | 885 | 14 |
| 1.5 | 3.2 | 75 | 912 | 844 | 786 | 847 | 51 |
| 1.5 | 3.2 | 80 | 726 | 777 | 705 | 736 | 30 |
| 1.5 | 3.2 | 85 | 692 | 709 | 843 | 748 | 68 |

Similar to the welding energy, the tensile strength is a response of the DoE. Therefore, a Pareto chart can again be constructed to evaluate the effects of the input parameters on this response. This chart is shown in Figure 6. From this chart, it is clear that—in descending order—the welding time, the interaction of welding time and pressure and amplitude, the interaction of welding time and pressure, and the vibration amplitude have statistically

significant effects on the weld strength. The pressure and the other interactions between the parameters do not have a statistically significant effect on the weld strength. According to these observations, the weld strength should not significantly increase or decrease for different levels of pressure when the other parameters are kept constant. Equation (4) shows the regression equation corresponding to this Pareto chart:

$$S = 1.600 - 1.018 \, WT - 0.504 \, P - 0.01958 \, A + 0.358 \, WT \times P$$
$$+ 0.01320 \, WT \times A + 0.00663 \, P \times A - 0.00470 \, WT \times P \times A \tag{4}$$

where S represents the tensile strength (in Mpa), WT the welding time (in s), P the pressure (in bar) and A the vibration amplitude (in %).

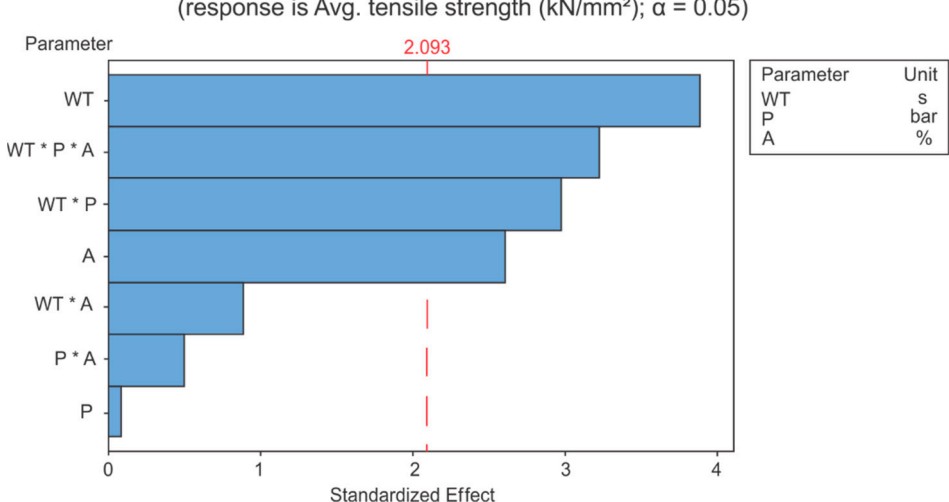

**Figure 6.** Pareto chart of the standardized effects of the welding parameters and their interactions on the average tensile strength of Al–Cu welds.

In the remainder of this section, the average joint strength will be represented as a percentage of the tensile strength of the aluminium base material. All weld specimens failed in the Al base material during tensile testing, which makes the strength of this weakest base material a proper reference for the weld strength. The aluminium base material strength was measured by performing tensile tests on three plates with dimensions 51.0 mm × 9.0 mm × 1.0 mm and calculating the average of the resulting forces at failure. These specimen dimensions are based on the size of the parallel overlapping weld configuration. The average measured force to break the aluminium base material equals 1.024 kN. When dividing this value by the original cross-section area of 9.0 mm$^2$, a tensile strength of 0.114 kN/(mm$^2$) = 114 Mpa is obtained.

The relation between the average tensile strength and the welding parameters can be seen in the contour plots in Figure 7. These plots show a very different behaviour for the welding times of 0.8 s, 1.5 s and 2.2 s. This means that there is no clear relation between the parameters individually and the weld strength, but optimization of the combination of parameters is required when searching for the maximal weld strength. For the shortest welding time, a low pressure value requires a low vibration amplitude value to obtain a weld strength above 90% of the base material. Higher pressure values allow the entire range of amplitude values and will still result in high weld strength. For an intermediate welding time, a very distinctive window of optimal pressure values can be observed. To obtain the maximal weld strength, however, the pressure and vibration amplitude should be chosen at the lowest levels of the parameter window. The maximal welding time shows two regions of maximal weld strength. The range of strengths that is achieved for this welding time is larger than for the other welding times, proving that when welding times are too high, it

becomes more difficult to realize a joint with maximal weld strength. This is in agreement with the observations of Liu et al., where the tensile strength of Cu–Al joints first increased and then decreased again with welding time at a constant amplitude, showing that there is an optimal welding time where a maximal weld strength is achieved [32].

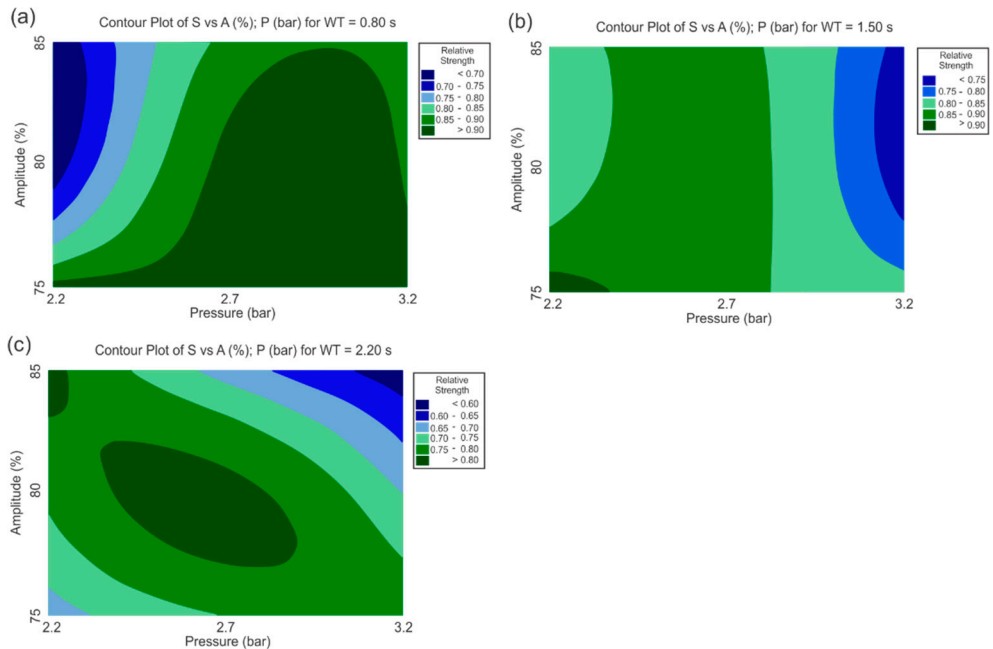

**Figure 7.** Contour plots of relative average tensile strength vs amplitude and pressure at constant welding time of (**a**) 0.8 s, (**b**) 1.5 s and (**c**) 2.2 s.

### 3.3. Relationship between Welding Energy and Tensile Strength

It can be expected that a higher energy input will result in a stronger weld since higher energy will result in a larger welded area. The downside of high welding energy is a large amount of heat introduced in the weld interface. This heat input will cause deformation of the material and, in the case of very soft aluminium plates, sticking of the plates to the anvil. Moreover, according to Kumar [33], the joint strength correlates strongly with the temperature developed at the interface. Therefore, it is to be expected that the maximal weld strength will be found for an optimized energy input. This input should be high enough to create a large welded area, but not too high, to avoid unwanted side-effects such as deformations and sticking.

In Figure 8 and Table 5, the average relative tensile strength with respect to the aluminium base material is plotted against the average welding energy. All strength measurements are below 1, meaning that although the welds failed in the aluminium base material, the joint strength is lower than the strength of the base material. This implies a decrease of the mechanical properties of the material during the weld cycle. The strength of the base material has decreased due to the heat input during the ultrasonic welding process. From the graph, there is no unambiguous relation between both output parameters. The measurements are scattered over a large range of weld strengths. One could argue that when the two datapoints at the bottom left are disregarded, the tensile strength decreases with increasing welding energy. This can then be explained by the observations above; an excessive heat input will cause deformations of the workpieces. All energy will then be used to accomplish these deformations, and less energy will be available to form a joint between the metal plates. The scattered positions of the measurements prove the sensitive nature of this joining process, as there is no easy relation that provides the required amount of energy to obtain the maximal weld strength. It remains important to evaluate different welding conditions and to determine the optimal combination of parameters to obtain an optimal weld strength.

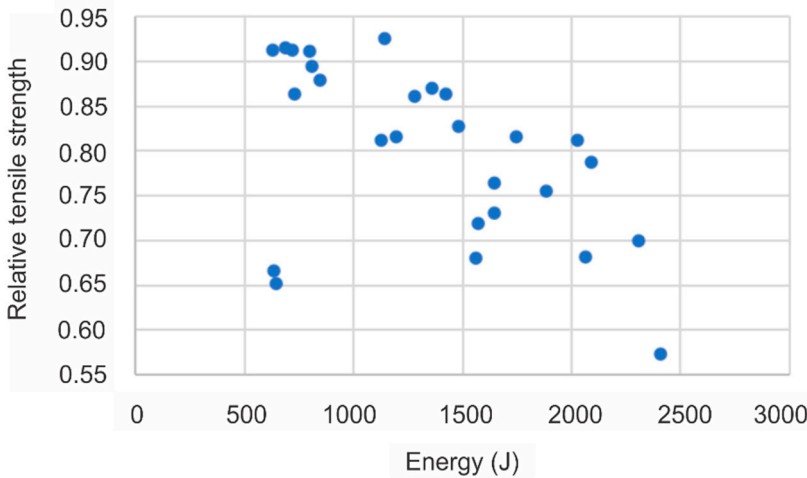

**Figure 8.** Relative tensile strength vs. average welding energy of Al–Cu welds.

**Table 5.** Influence of the welding parameters on the relative tensile strength. The background color indicates the correlation of the value with the DoE level (−1;0;1).

| Weld Time (s) | Pressure (bar) | Amplitude (%) | Relative Tensile Strength |
|---|---|---|---|
| 0.8 | 2.2 | 75 | 0.91 |
| 0.8 | 2.2 | 80 | 0.67 |
| 0.8 | 2.2 | 85 | 0.65 |
| 0.8 | 2.7 | 75 | 0.92 |
| 0.8 | 2.7 | 80 | 0.91 |
| 0.8 | 2.7 | 85 | 0.86 |
| 0.8 | 3.2 | 75 | 0.91 |
| 0.8 | 3.2 | 80 | 0.90 |
| 0.8 | 3.2 | 85 | 0.88 |
| 2.2 | 2.2 | 75 | 0.68 |
| 2.2 | 2.2 | 80 | 0.76 |
| 2.2 | 2.2 | 85 | 0.82 |
| 2.2 | 2.7 | 75 | 0.75 |
| 2.2 | 2.7 | 80 | 0.81 |
| 2.2 | 2.7 | 85 | 0.68 |
| 2.2 | 3.2 | 75 | 0.79 |
| 2.2 | 3.2 | 80 | 0.70 |
| 2.2 | 3.2 | 85 | 0.57 |
| 1.5 | 2.2 | 75 | 0.93 |
| 1.5 | 2.2 | 80 | 0.81 |
| 1.5 | 2.2 | 85 | 0.82 |
| 1.5 | 2.7 | 75 | 0.86 |
| 1.5 | 2.7 | 80 | 0.87 |
| 1.5 | 2.7 | 85 | 0.86 |
| 1.5 | 3.2 | 75 | 0.83 |
| 1.5 | 3.2 | 80 | 0.72 |
| 1.5 | 3.2 | 85 | 0.73 |

The effects of increasing the values of each parameter on the weld interface properties are summarized in Table 6.

**Table 6.** Effects of increasing parameter values on the weld interface.

| Parameter | Welding Time | Pressure | Vibration Amplitude |
|---|---|---|---|
| **Effects of Increasing Parameter Value** | Increased Welded Area | Deformation and Elongation of the Plates | Lowest Influence |
| | Wavy surface of Al Plate | Wavy surface of the Al Plate | |

### 3.4. Metallographic Examination

All welding conditions resulting in a weld with a peel test score of 3 or 5 were evaluated by metallographic examination. The welds with a peel test score of 1 could be completely separated and did not result in a welded interface. The classification of the welds based on the peel test score is shown in Table 7.

**Table 7.** Peel test score of the welds. The background color indicates the correlation of the value with the DoE level (−1;0;1).

| Weld Time (s) | Pressure (bar) | Amplitude (%) | Peel Test Score |
|---|---|---|---|
| 0.8 | 2.2 | 75 | 5 |
| 0.8 | 2.2 | 80 | 5 |
| 0.8 | 2.2 | 85 | 5 |
| 0.8 | 2.7 | 75 | 5 |
| 0.8 | 2.7 | 80 | 5 |
| 0.8 | 2.7 | 85 | 5 |
| 0.8 | 3.2 | 75 | 5 |
| 0.8 | 3.2 | 80 | 5 |
| 0.8 | 3.2 | 85 | 5 |
| 2.2 | 2.2 | 75 | 5 |
| 2.2 | 2.2 | 80 | 5 |
| 2.2 | 2.2 | 85 | 5 |
| 2.2 | 2.7 | 75 | 5 |
| 2.2 | 2.7 | 80 | 5 |
| 2.2 | 2.7 | 85 | 1 |
| 2.2 | 3.2 | 75 | 1 |
| 2.2 | 3.2 | 80 | 5 |
| 2.2 | 3.2 | 85 | 1 |
| 1.5 | 2.2 | 75 | 3 |
| 1.5 | 2.2 | 80 | 5 |
| 1.5 | 2.2 | 85 | 5 |
| 1.5 | 2.7 | 75 | 3 |
| 1.5 | 2.7 | 80 | 5 |
| 1.5 | 2.7 | 85 | 5 |
| 1.5 | 3.2 | 75 | 5 |
| 1.5 | 3.2 | 80 | 5 |
| 1.5 | 3.2 | 85 | 5 |



Figure 9 shows a detailed image of the weld interface of a Cu–Al weld with a peel test score of 5. On this image, it is clear that there is no mixing between the materials, and no welded islands can be distinguished (continuously welded interface). This observation deviates from the observations found in the work of M. P. Satpathy et al. [34], where small micro-joints can be distinguished for short welding times, followed by a material flow and bonding between the copper and aluminium plates for longer welding times. For this weld specimen, there is no change in microstructure in the proximity of the weld interface compared to the material further away from the interface. Due to the lack of the mixing of the materials and the absence of any welded islands, the determination of the length of the welded area is not a simple task.

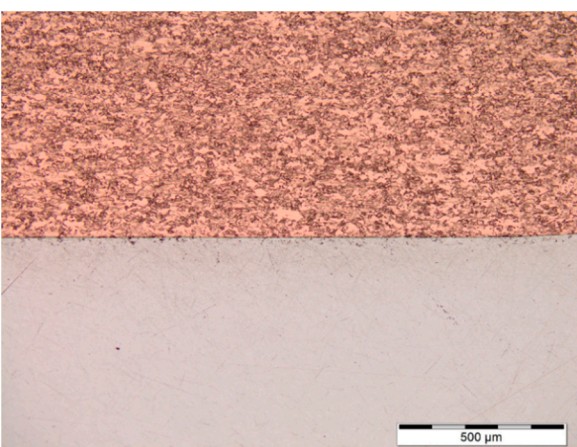

**Figure 9.** Detailed image (100×) of Al-Cu weld interface.

In Figure 10, the cross section of a standard high-strength weld is shown. The polished sample image in Figure 10a shows a flawless connection between the copper and aluminium plates. There is no black line present that separates the surfaces, and no weld flaws such as porosities can be distinguished. When examining the etched cross-section in Figure 10b, there is a clear distinction between the copper and aluminium plates. This observation can be made for most welded specimens; an extremely thin line can be observed on the etched cross-sections of the welds. This brings the question of whether the plates are connected at a metallurgical level or whether they are simply pressed very closely together. This question is not answered when enlarging the weld interface. The detailed image in Figure 10c does not show a line between the plates, but it also does not provide a decisive answer to the question if there is welded bond between the plates or not. Moreover, the image in Figure 10d shows a very thin black line between the metal plates, which could indicate a very thin gap. In addition to that, the microstructures in different regions were evaluated, indicating identical microstructures and grain sizes.

To evaluate if there is an infinitesimally thin gap between the metal plates, a more thorough investigation of the weld interface was conducted using SEM. In Figure 11, the SEM analysis of the specimen shown in Figure 10 is illustrated. The images show no signs of a gap between the plates. The images in Figure 11c,d show a bright white line at the weld interface. This indicates that the phase of the material has changed at this location. This is most likely the result of a large heat input at the weld interface. Phase transformations as a result of the heat input have been observed before by A. O'Brien et al. [35]. If an oxide layer had been formed on top of the Al or Cu plates at the weld interface, this oxide layer would have been broken down during the welding process as a result of the pressure and vibrations applied by the sonotrode [10].

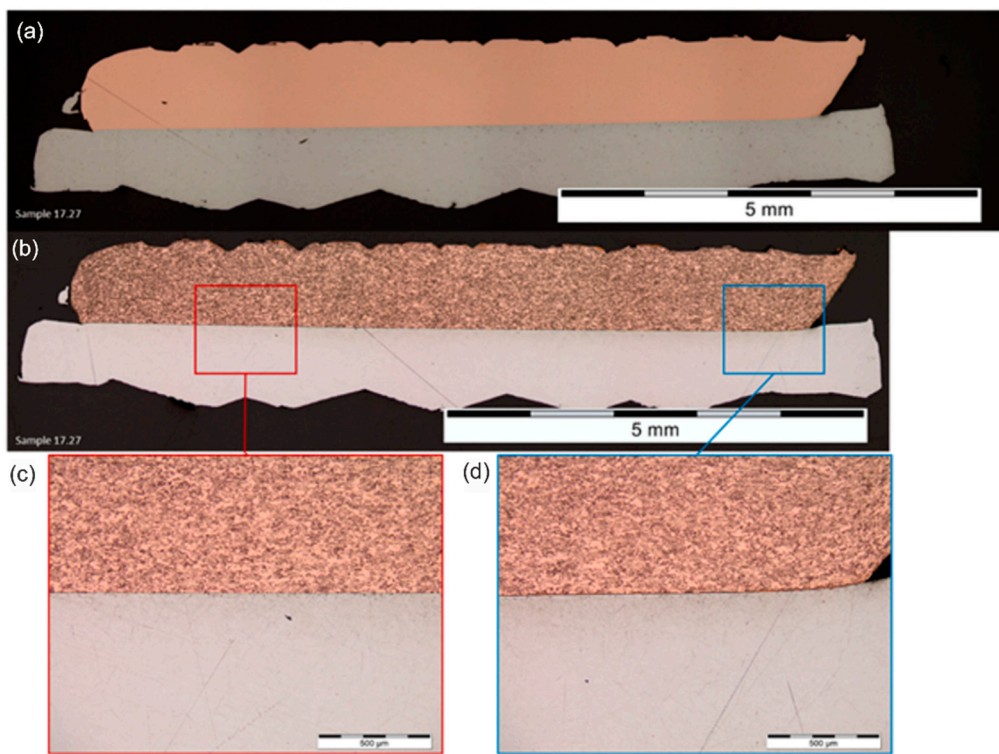

**Figure 10.** Cross section of a welded Al–Cu sample: (**a**) polished (50×); (**b**) etched (50×); (**c**,**d**) detailed (100×).

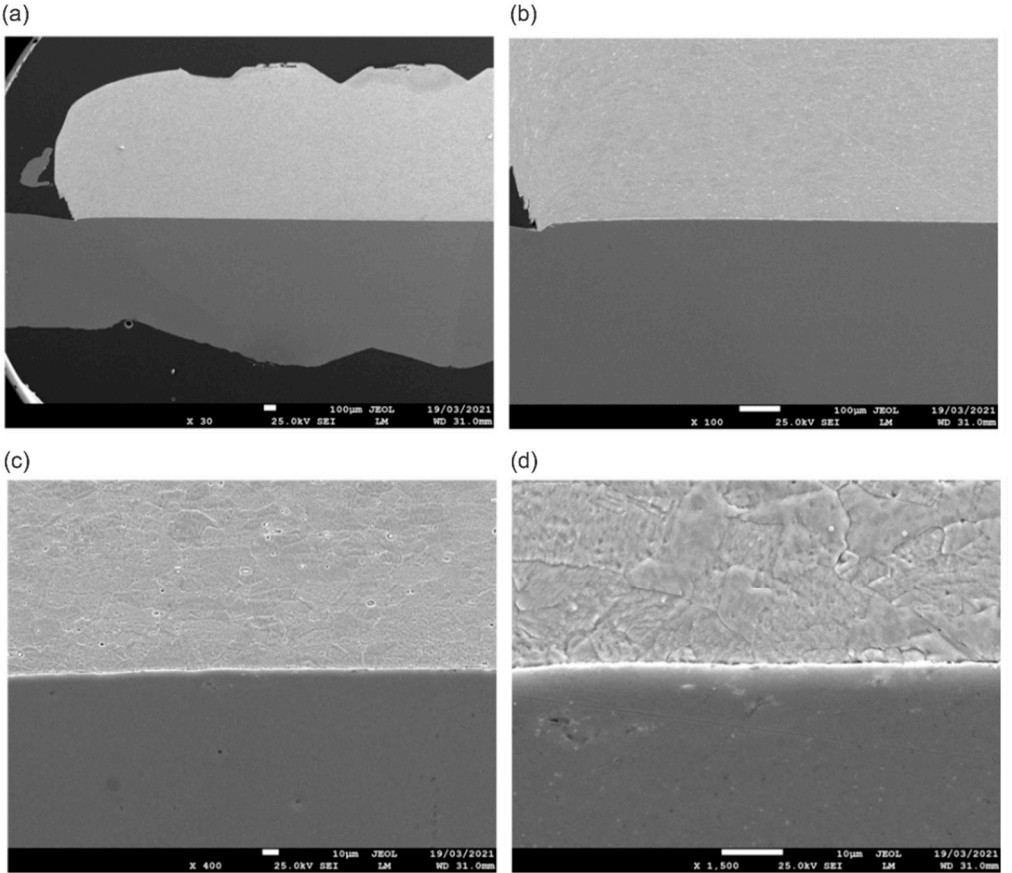

**Figure 11.** SEM images of the cross sections of an Al–Cu weld: (**a**) 30×; (**b**) 100×; (**c**) 400×; (**d**) 1500×.

To evaluate the potential presence of an oxide layer at the weld interface, a line scan is made using EDX. This scan determines the presence of different elements along a line perpendicular to the weld interface. Figure 12 shows the concentration of the Cu, Al and O at each point along the line. Here, cps stands for counts per second, referring to the X-ray count rate of the EDX process. The concentration of oxygen is almost negligible over the entire length of the line scan. There is no peak in the oxygen concentration at the location of the weld interface. This means that no oxide layer is present on top of the metal plate surfaces. In both line scans, the lines representing the Cu and Al overlap in the region of the weld interface. This indicates that there is a small amount of mixing between both materials, and a metallurgical joint is formed. The plates are thus not only pressed together, but there is a weld between the dissimilar metals.

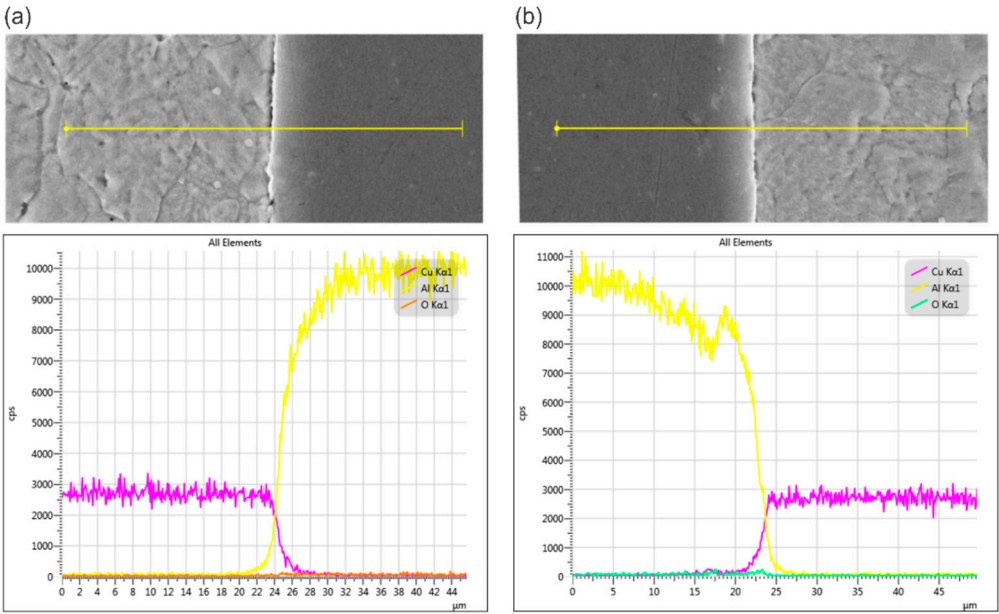

**Figure 12.** EDX line scan (1500×) of the Al–Cu weld interface for two different samples (**a**,**b**).

The EDX mappings in Figure 13 show a map of the elements present over a part of the weld cross-section. From this image, it is again clear that no concentration of oxygen at the weld interface was detected. There is a small amount of oxygen present inside the weld cross-section, but it is evenly dispersed in both metal plates. This low oxygen concentration is a residue from the breakdown of the oxide layer that occurred during the solid-state welding process. On the images in Figure 13, the mixing between the copper and aluminium plates is not as clear. The mixing is thus very limited, as it is only visible on the images at a magnification of 1500× and not on images with a magnification of 400×.

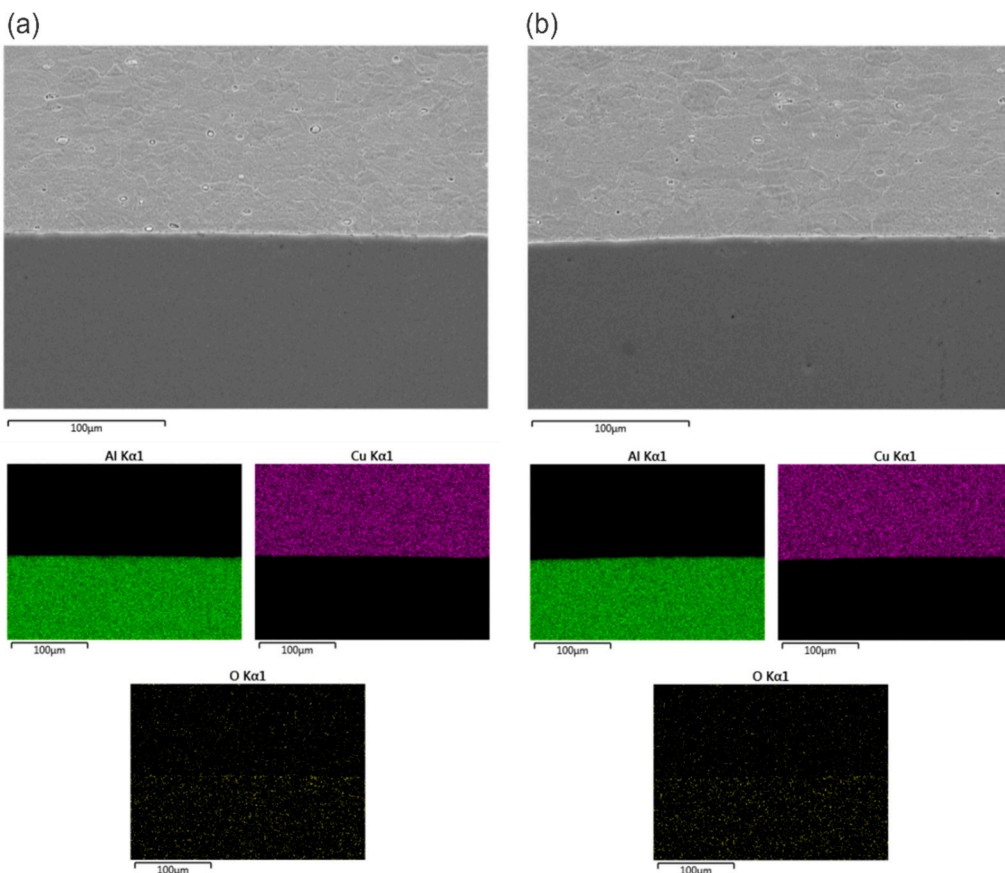

**Figure 13.** EDX mapping (400×) of Cu, Al and O content at the weld interface of an Al–Cu weld for two different samples (**a**,**b**).

## 4. Conclusions

The present work proposed a strategy to determine the window of welding parameters for ultrasonic welding of Al–Cu dissimilar metal joints. The influence of each parameter and their combinations on the welding energy and tensile strength of the lap shear specimen was evaluated. Furthermore, a metallographic investigation was performed for these welds.

Based on the results achieved, one can conclude:

- The welding time is the parameter with the most important influence on the welding energy and tensile strength.
- Outside the stipulated window of parameters, a large welding time will result in an increase of the heat input and, therefore, a decrease of the weld strength.
- There is a decrease of the mechanical properties during the weld cycle due to the heat input resulting from the friction at the weld interface. This can be concluded based on the observation that the relative tensile strength for all welding conditions is below 1.
- The SEM analysis shows no weld flaws or gaps between the plates, indicating a satisfactory joining of the dissimilar materials.
- Through EDX line scan and mapping, it can be concluded that no oxide layer or gap is present between the dissimilar metals, and there is some overlap between the aluminium and copper concentrations at the weld interface, indicating mixing between the plates.
- For the specific application of ultrasonic welding for 1.0 mm thick aluminium and copper sheets, it is recommended to use a welding time of 1.5 s, pressure of 2.2 bar and amplitude of 75% (49.5 μm) to achieve high strength.

**Author Contributions:** Conceptualisation, R.G.N.S. and K.F.; Data curation, R.G.N.S., S.D.M., K.F. and W.D.W.; Formal analysis, S.D.M.; Funding acquisition, K.F.; Investigation, R.G.N.S., S.D.M., K.F. and W.D.W.; Methodology, R.G.N.S., S.D.M., K.F. and W.D.W.; Project administration, K.F. and W.D.W.; Resources, K.F.; Software, R.G.N.S. and S.D.M.; Supervision, R.G.N.S., K.F. and W.D.W.; Validation, S.D.M.; Visualisation, R.G.N.S., S.D.M., K.F. and W.D.W.; Writing—original draft, R.G.N.S. and S.D.M.; Writing—review and editing, K.F. All authors have read and agreed to the published version of the manuscript.

**Funding:** This research received no external funding.

**Conflicts of Interest:** The authors declare no conflict of interest. The funders had no role in the design of the study; in the collection, analyses, or interpretation of data; in the writing of the manuscript, or in the decision to publish the results.

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
