# Peer review of "Development and Evaluation of the Ultrasonic Welding Process for Copper-Aluminium Dissimilar Welding†"

_jmmp, doi:10.3390/jmmp6010006_

Round 1
Reviewer 1 Report
This paper delivered a common experimental study on USW joining of Al to Cu. The quality of this paper is not good enough to published for the following reasons:
- The literature research about USW of Al to Cu is insufficient, indicating that the author did not read the relevant publications carefully;
- How did the data in the equations come from is not clear;
- The quality of the metallographic figures is so poor that the readers are hard to obtain effective information;
- The conclusions are too long, and bad in organization, which indicates that the authors did not summarize the content of this paper well;
- The content of this paper is lack of innovation.
Author Response
The comments and replies to all of the comments from the reviewer can be found in the attached file.
In name of the authors, I would like to thank the comments and contribution of the reviewer.
We are confident that the level of the article was improved after the review.

Reviewer 2 Report
Dear Authors,
the manuscript «Development and Evaluation of the Ultrasonic Welding Process of Copper-Aluminium Dissimilar Welding» has relevance and novelty for the automotive and electrical industries.
However, the content of the manuscript could be improved according to the comments below:
- The data in tables 3, 4 would be better presented in the form of diagrams or graphs.
- Please describe the process of preparing the surface of the samples before welding.
- The manuscript states (364-365): «From this image, it is again clear that there is no concentration of oxygen at the weld interface. It may be more correct to write «not detected»?
- And (365-368) «There is a small amount of oxygen present inside the weld cross-section, but it is evenly dispersed in both metal plates. This oxygen most likely originates from the base material and is not the result of an oxide reaction due to the heat development during the welding process». Please clarify the presence / absence of oxide films of Al2O3 at the interface and show interface microstructure at high magnification (if possible).
- We recommend expanding references to 20-30, accordingly changing the introduction, and replacing Internet links with scientific articles.
- Please correct the abstract and especially the conclusions, include significant experimental characteristics, including optimal welding parameters to achieve high strength.
Best regards,
Reviewer
Author Response

(The authors gave the same response as above.)

Reviewer 3 Report
Review of the manuscript: Development and Evaluation of the Ultrasonic Welding Process of Copper-Aluminium Dissimilar Welding
Dear Authors,
The work concerns the evaluation of ultrasonic dissimilar welds. Overall, it may be interesting despite some concerns.
Something went wrong with the template.. two tables ( 1 and 2) are unreadable, the same for the journal logo.
Line 32: why did you define Cu as a lightweight material? The specific weight of Cu is 8.96 g/cm3 (higher than Fe)... Al, Ti and Mg are lightweight. The entire introduction paragraph must be improved. Non enough references are included; furthermore, more applications examples of state of the art must be included. Traditional techniques' defects in joining materials are various; maybe some examples can be reported to highlight why you preferred the ultrasonic process. There are multiple manuscripts concerned with the defects in Al (or Cu) welded parts in literature.
Line 82: vale= value
Line 88: Table 1 is not readable; I can read only the alloy elements name (Al, Si, Fe, Cu, Mn, Mg, Ti).
Line 130: why the classification was Score 1, 3 and 5 and not 1,2 and 3? It must be explained.
The DOE was unreadable. I can only read 'doe level, welding time (s), pressure' and the lines -1 and 0.
Are the ultrasonic welding and tensile machines pictures helpful in this manuscript? I think they are unnecessary.
Line 167: Please explain the means of Energy A, B, C and D.
Line 212: why did you start with Tensile B and not A? Moreover, in my opinion, the values should be reported in MPa instead of kN. Furthermore, kN is unnecessary; the values are pretty low. Consider changing the unit of measurement in MPa or at least N.
Line 241: is the conversion from kN to MPa necessary? The value in MPa is ok.
Line 260: reference number [14] must be reported near 'Liu et al.' in line 257.
Line 271: reference number [15] must be reported near 'Kumar' in line 270.
Line 315:may you show the microstructures of both materials further away from the interfaces? Did you notice the identical microstructures (i.e. grain size)?
Line 320: the polished image= the polished sample image? <why did you only etch the Cu and not Al?
Figure 12: Images in Figure 12 appears unnecessary since you reported the SEM images refuting the previous considerations about the possible air gap present. The entire discussion about these images seems useless.
You performed an EDX line analysis, but you could do a point analysis or a short area analysis into the white film to understand its real composition. What phase is it? In the end, we know about the thin white film existence but we do not know what it is?
Conclusions are in line with the manuscript.
Reference 5 must be checked; it was reported XXXX instead of the manuscript number.
I suggest to the Authors to check the English language and to enhance the introduction paragraph. For these reasons, I suggest major revisions.
Regards,
Reviewer.
Author Response

(The authors gave the same response as above.)

Round 2
Reviewer 3 Report
Dear authors, thank you for improving the manuscript. The introduction was improved as the whole manuscript.
After the revision process, I recommend the acceptance in the present form.
Best regard